# On Effectiveness and Efficiency of Agentic Tool-calling and RL Training

**Tong Liu**[* 1]  **Cheng Qian**[2]  **Matej Cief**[3]  **Yuan He**[3]  **Daniele Dan**[3]  **Nikolaos Aletras**[* 4]  **Gabriella Kazai**[3]

## Abstract

Tool-calling is a central component of modern large language model (LLM) agents, equipping them with skills beyond their parametric knowledge. This paper studies tool-calling along two complementary axes: **effectiveness**, i.e., how this capability is *measured*, and **efficiency**, i.e., how it is *learned*. On effectiveness, we systematically analyze tool-calling evaluation pipelines and show that results can be highly sensitive to seemingly minor, often undocumented implementation choices including the random seed, system prompt, multi-turn template construction, and how prior interaction/reasoning history is carried forward. These choices can lead to substantial differences in reported performance, especially in multi-turn settings where without rigorous standardization, leaderboard rankings are unreliable. On efficiency, we examine standard reinforcement learning (RL) for tool-calling and identify two sources of computational waste: (i) during rollouts, many prompts produce no learning signal, and (ii) during policy updates, optimization incurs high computational cost. Guided by these findings, we introduce two techniques that accelerate RL-based tool-calling training, achieving substantial wall-clock speedup without degrading performance.

## 1. Introduction

*Tool-calling* (or function calling) has become a cornerstone of recent progress in LLM agents (Openai, 2025; Anthropic, 2025b; Deepmind, 2025; xAI, 2025; Meta, 2025). By interacting with external resources, e.g., calling APIs, agents can perform tasks that would otherwise be difficult or impossible to solve by solely replying on LLM parametric knowledge. Consequently, the community has adopted standardized tool-calling benchmarks such as BFCL (Patil et al., 2025) and Tau-series (Barres et al., 2025; Shi et al., 2026), and post-training methods such as Reinforcement Learning (RL) to improve the accuracy and robustness of tool calls.

However, the *effectiveness* of agentic tool-calling is only as credible as the evaluations used to measure it, and evaluation quality remains a critical yet under-examined issue. When benchmark results are unreliable, the field may chase superficial gains while overlooking genuinely promising approaches (Dehghani et al., 2021; Henderson et al., 2018; Liao et al., 2021). While reproducibility and benchmark sensitivity have been studied in other areas of foundation models (Reuel et al., 2024; Biderman et al., 2024; Hochlehnert et al., 2025), such analysis for tool-calling is unexplored. Therefore, we extensively scrutinize the evaluation pipeline that underpins the effectiveness of agentic tool-calling. Using the BFCL (Patil et al., 2025) benchmark as a case study, *we show that reported tool-calling performance can swing substantially due to seemingly minor, yet often undocumented choices, including random seeds, multi-turn templates, history handling, and training-data influence. This sensitivity can substantially hinder meaningful comparisons unless it is carefully controlled.*

Having examined how fragile evaluation can distort measured effectiveness, we turn to *efficiency*: the computational cost of acquiring tool-calling capability. In particular, we identify major efficiency bottlenecks in RL-based tool-calling training. For common RL algorithms such as PPO (Schulman et al., 2017) and GRPO (Shao et al., 2024), training typically alternates between (i) *rollout generation*, where the policy samples tool-call trajectories for each prompt, and (ii) *policy updates*, where the model is optimized on the collected rollouts. *We find both stages are surprisingly inefficient: up to 80% of prompts produce rollouts that yield no gradient signal, and the update stage can dominate wall-clock time, costing 3–5× times as much computation as rollout generation.*

To reduce overhead in both stages, we propose two simple yet extremely effective accelerations. (1) **Online pre-rollout filtering:** before generating rollouts, we skip prompts whose rollouts were entirely correct in the previous $k$ epochs, avoiding redundant sampling. While related

---

[*]Work done while at Amazon. [1]LMU Munich & Munich Center for Machine Learning [2]UIUC [3]Amazon [4]University of Sheffield. Correspondence to: Tong Liu <tongliu.physics@gmail.com>, Gabriella Kazai <gkazai@amazon.co.uk>.

ideas have been explored for math reasoning (Zheng et al., 2025), we find that for tool-calling, excluding consistently-correct prompts for just one or two epochs is already effective (Fig. 5). (2) **Variance-aware rollout down-sampling:** motivated by the high update cost in tool-calling (Fig. 6), we adopt the rollout down-sampling strategy of Xu et al. (2025): updating the policy using only a subset of the generated rollouts, selected to maximize reward variance, thus substantially reducing policy-update computation.

Our contributions are summarized as follows:

- **Effectiveness:** We systematically study agentic tool-calling evaluation, using BFCL as a case study. We show that small differences in evaluation pipelines can lead to large performance differences, complicating reproducibility and cross-paper comparisons.
- **Efficiency:** We identify two major inefficiencies in RL-based tool-calling training and propose two effective techniques that deliver substantial end-to-end wall-clock speedups without sacrificing final performance.

**Conflict of Interest Disclosure.** The authors are employed by Amazon, which develops Nova-1, one of the models evaluated in this paper.

## 2. Preliminaries

### 2.1. Task formulation

For tool-calling task $i$, let $x_{i,1}$ denote the initial user query and $T_i = \{t_{i,1}, \ldots, t_{i,m}\}$ the corresponding available tools. We define the task-specific system prompt as $\mathrm{sys}_i := (\mathrm{sys}, T_i)$, where sys is a shared system prompt and $T_i$ encodes the tool schemas. Given a $k$-turn multi-turn conversation, we denote the trajectory prefix as an ordered sequence:

$$s_{i,k} = \langle \mathrm{sys}_i, (x_{i,1}, y_{i,1}, o_{i,1}), \ldots, (x_{i,k}, y_{i,k}, o_{i,k}) \rangle, \quad (1)$$

where $x_{i,h}$, $y_{i,h}$, and $o_{i,h}$ are the user query, model response, and environment observation (e.g., tool outputs) at turn $h$. The response $y_{i,h}$ may invoke a subset of tools $T_{i,h} \subseteq T_i$. Depending on the conversation, $x_{i,h}$ and $o_{i,h}$ may be absent. The goal of tool-calling is to generate a proper $y_{i,h}$ that correctly invoke tools when needed and effectively address the corresponding user queries.

### 2.2. RL formulation

Under GRPO, given $s_{i,k}$ the policy $\pi_\theta$ samples $n$ rollouts $\{y_{i,k+1,1}, \ldots, y_{i,k+1,n}\}$ and receives verified rewards $\{r_{i,k+1,1}, \ldots, r_{i,k+1,n}\}$. After group normalization, the advantage for rollout $j$ is

$$A_{i,k+1,j} = \frac{r_{i,k+1,j} - \bar{r}_{i,k+1}}{\sigma_{i,k+1}}, \quad (2)$$

where $\bar{r}_{i,k+1}$ and $\sigma_{i,k+1}$ are the within-group mean and standard deviation. $A_{i,k+1,j}$ participates in the gradient of the objective regarding the sample $s_{i,k}$. The objective can be written as:

$$\mathcal{L}_{\mathrm{GRPO}}(\theta) = \mathbb{E}_{(s_{i,k}, y_{i,k+1,j}) \sim (\mathcal{D}, \pi_\theta)}[\min(\rho_{i,k+1,j} A_{i,k+1,j},$$
$$\mathrm{clip}(\rho_{i,k+1,j}, 1 - \epsilon, 1 + \epsilon) A_{i,k+1,j})], \quad (3)$$

where $\rho_{i,k+1,j} = \frac{\pi_\theta(y_{i,k+1,j}|s_{i,k})}{\pi_{\mathrm{old}}(y_{i,k+1,j}|s_{i,k})}$ is the probability ratio between the current and old policies. Critically, when all rollouts within a group receive identical rewards, $A_{i,k+1,j} = 0$ for all $j$, resulting in zero gradient contribution. We refer to such prompts as *zero-variance prompts*.

### 2.3. Experimental Setup

**Models, datasets and hyperparameters.** On effectiveness, we run five commonly used models, three Qwen-series models, Qwen3-4B, Qwen3-8B, Qwen2.5-7B-Instruct, and two Llama-series models, Llama3.1-8B-Instruct and Llama3.2-3B-Instruct. For the training part, We train two representative settings: Qwen2.5-3B-Instruct for single-turn tool-calling and Qwen3-4B (Yang et al., 2025) for multi-turn tool-calling. More details about models refers to Appx C. Our single-turn training set is built from xLAM (Zhang et al., 2025a) and a subset of ToolACE (Liu et al., 2024), following the preprocessing in Zhang et al. (2025c) with an additional policy-based filter to remove overly easy or overly hard prompts. Our multi-turn training set is constructed from Zhang et al. (2025b) with further extraction, cleaning, and filtering. Dataset statistics are reported in Table 1, and full preprocessing details are deferred to the Appx D. All RL experiments are implemented in the `VERL` framework (Sheng et al., 2024). Also see Appx F for hyper-parameters of training.

*Table 1.* Statistics of single-turn and multi-turn data.

|  | Single-Turn | Multi-Turn |
|---|---|---|
| Raw Data | 63k | 23k |
| After Processing | 2.3k | 2.6k |

**Benchmark and evaluation** We evaluate models' tool-calling performance using the BFCL benchmark in Section 3. BFCL is a widely used benchmark covering a diverse set of APIs. It primarily includes categories of Single-turn Non-Live (synthetically generated single-turn data), Single-turn Live (real user-contributed single-turn data) and Multi-turn tasks. In Section 4, we also report the evaluation on ACEBench (Chen et al., 2025), focusing on the English data across all categories. We employ Claude 4 as the user simulator during evaluation. We observe occasional role drift (assistant-like replies), we append a single constraint sentence to the simulator instructions: "You must respond as a USER, not as an assistant".

We compare against closed-source baselines (Claude Sonnet 4 (Anthropic, 2025a), Nova 1 Lite/Pro (Amazon Artificial General Intelligence, 2024)) and open-source models (Gemma3-Instruct 27B (Team et al., 2025), Magistral-Small-2509 24B (MistralAI, 2025)).

## 3. Effectiveness Under the Microscope: How Fragile is Tool-calling Evaluation?

Tool-calling benchmarks are widely used to quantify agent *effectiveness*, yet the evaluation pipeline itself introduces many unexamined degrees of freedom. We use BFCL (Patil et al., 2025) as a case study to examine whether seemingly innocuous implementation choices can substantially change reported performance, especially in multi-turn settings where small deviations compound across steps. Our goal here is not to "optimize" benchmark scores, but to identify *sensitivity points* that must be controlled (or at least reported) for meaningful comparisons. In our study, we consider random seeds, multi-turn template construction, reasoning history, system prompts and the effect of training data.

### 3.1. Random Seed Variance

Prior work has shown that deep RL algorithms are highly sensitive to random seeds (Henderson et al., 2018; Chan et al., 2019; Colas et al., 2018). However, this factor is often overlooked in tool-calling literature that typically reports results on a single run (e.g., xAI (2025); Yang et al. (2025); Qian et al. (2025); Zhang et al. (2025c)). Concretely, we investigate this by evaluating BFCL (Patil et al., 2025) runs under 10 random seeds for five commonly used models, i.e., Qwen-series and Llama-series ones.

Fig. 1 summarizes the results: Single-turn performance is relatively stable, but multi-turn scenarios exhibit notably higher variance: early stochastic differences can alter subsequent tool calls and push the interaction onto divergent trajectories.

> **Takeaway 3.1** Single-turn BFCL is largely stable across different random seeds, but multi-turn evaluation is noticeably more seed-sensitive (up to ∼3% deviation), since small early deviations can compound over turns.

From this point on, we report BFCl results averaged over three random seeds, unless otherwise stated.

### 3.2. Multi-turn Template Variance: Native vs. Context

A second, frequently under-documented factor is the construction of the multi-turn template. As illustrated in Fig. 2, the "native" approach represents history as role–content messages that are later formatted by the official chat template, whereas the "context" approach injects the entire dialogue history (including intermediate reasoning and tool I/O) into a single user turn, as done in some prior work (e.g., Qian et al. (2025)). Although both choices appear superficially similar (as they convey the same information), they induce different formatting and tokenization, and therefore different behavior.

Fig. 3 left shows that using the native multi-turn template yields a consistent ∼6–8% gain over the context template across three models, Qwen3-8B, Qwen3-4B and Qwen2.5-7B-Instruct. This highlights a key implication: multi-turn tool-calling accuracy is not solely a property of the model, but also of *how* the interaction history is serialized.

> **Takeaway 3.2** Multi-turn template construction can substantially affect tool-calling effectiveness: "native" serialization outperforms "context" concatenation by ∼6–8% on BFCL multi-turn.

Going forward, we use the model's native multi-turn template for both training and evaluation.

### 3.3. Thinking History Variance

Another under-explored factor is whether intermediate reasoning traces are retained across turns. Reasoning content (e.g., chain-of-thought or `<think>` blocks) can dominate the context budget in multi-turn interactions, forcing a practical trade-off between preserving reasoning history and conserving tokens. This choice also interacts with how multi-turn data is constructed (e.g., many datasets omit intermediate reasoning, such as Prabhakar et al. (2025)).

We compare multi-turn BFCL performance with and without thinking-history retention across Qwen variants. As shown in Fig. 3 middle, retaining thinking history consistently improves Qwen3 models (e.g., ∼3–5% for Qwen3-8B and Qwen3-4B), suggesting these models can leverage prior reasoning to maintain coherent tool-calling behavior.

> **Takeaway 3.3** Retaining thinking history improves by ∼2–5% on BFCL multi-turn for reasoning models.

From now on, we keep the full thinking history in the multi-turn template.

### 3.4. System Prompts Can Reshape the Baseline

A line of work has shown that benchmark-driven progress can be distorted by under-controlled experimental factors, such as weaker baselines; for example, Hochlehnert et al. (2025) report that results on math tasks can drop markedly under standardized evaluation. We highlight an analogous

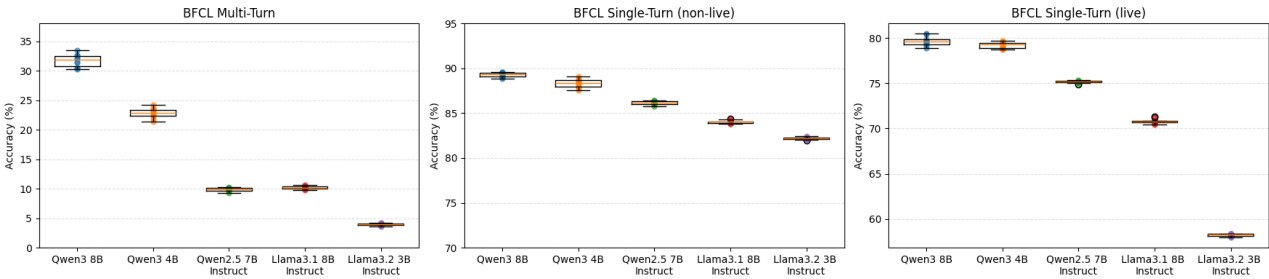

*Figure 1.* Tool-calling performance across ten different random seeds on BFCL.

```
Multiturn Template (Native)

<SYS>
SYSTEM_PROMPT
</SYS>
<USR>
USER_QUERY 1
</USR>
<AST>
<think>
THINKING PROCESS 1
</think>
TOOL-CALLING CONTENT
</AST>
<USR>
<tool_response>
TOOL-CALLING RESPONSE
</tool_response>
<tool_response>
TOOL-CALLING RESPONSE
</tool_response>
</USR>
<AST>
ASSISTANT RESPONSE
</AST>
<USR>
USER QUERY 2
</USR>
<AST>
```

```
Multiturn Template (Context)

<SYS>
SYSTEM_PROMPT
</SYS>
<USR>
Dialogue Records History
<user>
USER_QUERY 1
</user>
<think>
THINKING PROCESS 1
</think>
<assistant>
TOOL-CALLING CONTENT
</assistant>
<tool_response>
TOOL-CALLING RESPONSE
</tool_response>
<tool_response>
TOOL-CALLING RESPONSE
</tool_response>
<assistant>
ASSISTANT RESPONSE
</assistant>
<user>
USER QUERY 2
</user>
</USR>
<AST>
```

```
Multiturn Template
(w/o thinking history)

<SYS>
SYSTEM_PROMPT
</SYS>
<USR>
USER_QUERY 1
</USR>
<AST>
TOOL-CALLING CONTENT
</AST>
<USR>
<tool_response>
TOOL-CALLING RESPONSE
</tool_response>
<tool_response>
TOOL-CALLING RESPONSE
</tool_response>
</USR>
<AST>
ASSISTANT RESPONSE
</AST>
<USR>
USER QUERY 2
</USR>
<AST>
```

*Figure 2.* Left: Native template. Middle: Context template. Right: Template without thinking history. We use abstract role markers (e.g., <SYS>, <USR>, <AST>) to represent model-specific chat-template tokens such as <|im_start|>system in Qwen-series models and <|start_header_id|>system in Llama.

risk for tool-calling: evaluation results may be misleading when models are evaluated using different system prompts, while prompt details are often treated as inconsequential.

We test this by making a small, manual modification to the default BFCL system prompt[1]. Specifically, we add a few instructions tailored for multi-turn interactions and thus intended to be *stronger* for multi-turn tool-calling (See Appx A). Fig. 3 right shows that this small change substantially improves BFCL multi-turn performance for Qwen3-4B and Qwen3-8B; notably, for Qwen3-4B the improvement is comparable to (or larger than) gains typically attributed to RL training (Section 4.3). This suggests that without prompt standardization, "method" improvements can be difficult to disentangle from prompt-induced performance shifts.

---

[1]We do not tune prompts with methods such as GEPA (Agrawal et al., 2025), but note that such prompt tuning could further amplify these effects.

> **Takeaway 3.4** Small system-prompt changes can produce gains that rival or exceed RL fine-tuning impact, underscoring the need to standardize (or at least report) prompts when comparing methods.

### 3.5. Effect of Training Data: Single-turn vs. Multi-turn

We next study how the *format* of tool-calling supervision affects performance. Multi-turn trajectories are substantially more expensive to collect and curate than single-turn examples, but it is unclear when multi-turn supervision is actually necessary and whether it transfers across settings.

Existing approaches typically mix a small amount of multi-turn data with predominantly single-turn data, without isolating their individual contributions. For example, ToolRL (Qian et al., 2025) includes ∼2.5% multi-turn examples (where tool-calling is required) in its 4k training

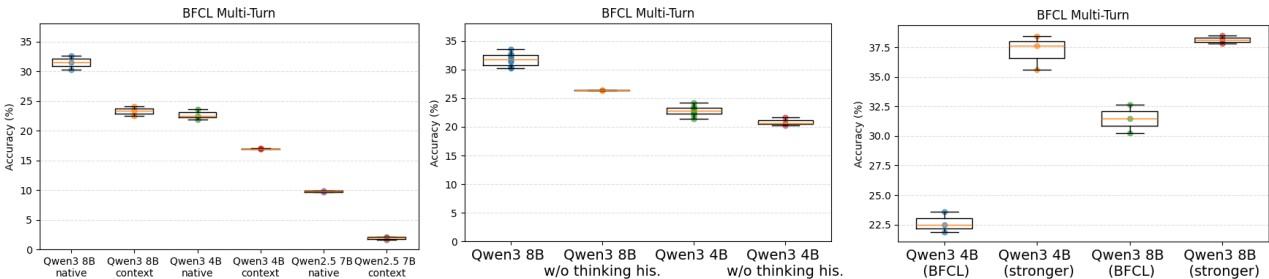

*Figure 3.* (Left): Influence of multi-turn templates on tool-calling performance for two Qwen models on BFCL multi-turn category. (Middle): Influence of retaining thinking history on tool-calling performance for two Qwen models on BFCL multi-turn category. (Right): Influence of system prompt on BFCL multi-turn category.

*Table 2.* Effect of training-data format on Qwen3-4B. Training results are measured in a single run. Multi-turn supervision does not improve multi-turn BFCL in this controlled setting, while single-turn training preserves multi-turn performance and slightly improves single-turn results.

| Qwen3-4B | BFCL Multi-turn | BFCL Single-turn | |
|---|---|---|---|
| | Multi-turn | Non-live | Live |
| **Base** | $22.7 \pm 0.9$ | $88.5 \pm 0.6$ | $79.3 \pm 0.4$ |
| Training on single-turn | $20.2 \pm 0.6$ | $89.5 \pm 0.5$ | $79.9 \pm 0.2$ |
| Training on multi-turn | $15.9 \pm 0.4$ | $88.6 \pm 0.4$ | $80.7 \pm 0.1$ |

set, while Tool-N1 (Zhang et al., 2025c) uses only ∼1.1% multi-turn data among 63k training examples.

We conduct a controlled experiment that isolates training format by fine-tuning separate models on *pure* single-turn vs. *pure* multi-turn data. We construct two matched training datasets derived from xLAM (Zhang et al., 2025a) and ToolACE (Liu et al., 2024) and finally both a multi-turn and a single-turn set with the same 0.7k size. See E for details.

**Findings.** Under this controlled data budget, multi-turn supervision does *not* reliably improve multi-turn BFCL; in fact, it degrades it, while yielding only marginal single-turn gains. In contrast, single-turn training improves single-turn BFCL and largely preserves the baseline multi-turn performance. A plausible explanation is that current multi-turn trajectories are a noisy training signal: errors and ambiguities accumulate over steps, and "correct" labels in earlier turns can encode suboptimal decisions that later derail the trajectory. Practically, this suggests that multi-turn tool-calling accuracy may be bottlenecked less by the *presence* of multi-turn data and more by the *quality and alignment* of trajectories. See Appx. B for a simple similarity measurement between data.

> **Takeaway 3.5** Multi-turn training does not automatically yield stronger multi-turn tool-calling performance; gains may be bottlenecked by trajectory quality and benchmark alignment rather than the amount of multi-turn supervision.

From this point on, we consider two complementary training settings. First, we focus on improving single-turn tool-calling by training on single-turn data, which is easier to scale and particularly effective for models with weaker single-turn performance, such as Qwen2.5 small models. Second, we target multi-turn tool-calling by training on multi-turn data, using curated, higher-quality trajectories to mitigate the noise and error accumulation observed in standard multi-turn supervision (see Section 2.3 for further details).

## 4. Efficiency Under the Hood: Where Does RL Tool-calling Training Waste Computation?

We now study the complementary axis of *efficiency*: the wall-clock cost of training tool-calling agents with RL. Using GRPO as a representative algorithm, we identify two dominant sources of wasted computation in practice and propose lightweight fixes that reduce end-to-end training time without sacrificing performance.

### 4.1. Issue 1: Rollout Waste from Zero-Variance Prompts

Tool-calling RL exhibits an unexpectedly high fraction of zero-variance prompts, cases where sampled rollouts provide no learning signal. Fig. 4 illustrates that this fraction can be large early in training and can change over time as the policy evolves; similar trends are observed for larger models, see Appx. G. This non-stationarity is important: prompts may move between "informative" and "uninformative" regimes as the model improves, making one-shot, pre-training filtering unreliable.

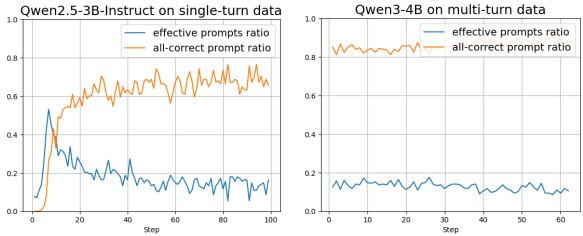

*Figure 4.* Ratio of zero-variance vs. non-zero-variance prompts during RL training of Qwen2.5-3B-Instruct on the single-turn tool-calling dataset and Qwen3-4B on the multi-turn data. Blue: ratio of prompts whose rollout rewards exhibit variance (i.e., useful learning signal). Orange: ratio of prompts with all rollouts achieving the maximum reward. In both cases, only around 20% prompts are effective prompts, indicating significant rollout waste.

**Method: online pre-rollout filtering.**   Motivated by the prevalence and non-stationarity of zero-variance prompts in tool-calling RL, we introduce an *online* pre-rollout filter that skips prompts unlikely to provide learning signal. The key challenge is that the set of "uninformative" prompts changes as the policy improves: a prompt that initially exhibits reward variance may later become uniformly solved (all-correct) or uniformly failed, and vice versa. This makes static, pre-training filtering unreliable. Instead, we estimate prompt usefulness *on the fly* using recent rollout outcomes.

Our central empirical observation is that *once a prompt becomes uniformly solved, it typically remains solved for many subsequent epochs*. Fig. 5 quantifies this temporal stability via $P(\text{still all-correct} \mid \text{conti-}k \text{ all-correct})$, the conditional probability that a prompt stays all-correct given it was all-correct for the previous $k$ consecutive epochs. Across most of training, this probability is already high with $k{=}1$ (exceeding 0.8 for single-turn and 0.9 for multi-turn), indicating that short-horizon history is a strong predictor of near-term redundancy. This enables a simple and conservative rule: before generating expensive rollouts, we skip prompts that have been all-correct for the past $k$ epochs.

**Implementation.**   We maintain a lightweight cache of per-prompt "all-correct" streaks and resample the active training set at the start of each epoch. Formally, given the original dataset $\mathcal{D}^{(0)}$, define an all-correct indicator for prompt $s_{i,k}$ at epoch $e$:

$$z_{i,k+1}^{(e)} = \mathbb{1}\left[ r_{i,k+1,1}^{(e)} = \cdots = r_{i,k+1,n}^{(e)} = r_{\max} \right], \quad (4)$$

where $\{r_{i,k+1,j}^{(e)}\}_{j=1}^{n}$ are rewards from $n$ rollouts at epoch $e$ and $r_{\max}$ is the maximum possible reward. We then update the consecutive all-correct count:

$$c_{i,k+1}^{(e)} = \begin{cases} c_{i,k+1}^{(e-1)} + 1 & \text{if } z_{i,k+1}^{(e-1)} = 1, \\ 0 & \text{otherwise.} \end{cases} \quad (5)$$

The resampled dataset at epoch $e$ is

$$\mathcal{D}^{(e)} = \left\{ s_{i,k+1} \in \mathcal{D}^{(e-1)} \; : \; c_{i,k+1}^{(e)} < k \right\}. \quad (6)$$

Only prompts in $\mathcal{D}^{(e)}$ receive rollouts at epoch $e$; prompts excluded by the filter are temporarily skipped. In practice, this reduces rollout computation by $|\mathcal{D}^{(e-1)} \setminus \mathcal{D}^{(e)}|$ while preserving informative prompts for learning.

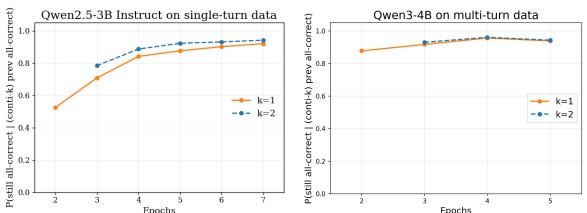

*Figure 5.* Temporal stability of all-correct prompts across training epochs. We plot $P(\text{still all-correct}\mid (\text{conti-}k) \text{ all-correct})$, the probability that a prompt remains to have all-correct rollouts given its rollouts were all-correct for the previous $k$ consecutive epochs. Both $k = 1$ and 2 exhibit high retention rates, demonstrating that recently-solved prompts exhibit strong temporal coherence and can be *safely* filtered to reduce redundant rollouts.

### 4.2. Issue 2: High Computation During Policy Update

RL algorithms such as GRPO benefit from using multiple rollouts per prompt to stabilize advantage estimates. In multi-turn tool-calling, however, increasing this $n$ quickly becomes *computationally prohibitive*. To understand why, we profile per-step wall-clock time under the VERL framework (Sheng et al., 2024).

Fig. 6 reveals a pronounced training-time asymmetry. First, policy updates grow much faster than rollout generation as $n$ increases, consistent with trends reported in math reasoning (Xu et al., 2025). Second, tool-calling amplifies this effect: unlike math, the policy update stage already **dominates** total runtime even at small $n$ (e.g., $n{=}4$). We attribute this to the substantially longer sequences in tool-calling, including tool schemas in the system prompt, multi-turn context, and tool I/O, which inflate the number of tokens backpropagated during updates.

**Method: max-variance rollout down-sampling.**   To reduce update cost while preserving the benefit of sampling diversity, we adopt *max-variance rollout down-sampling* (Xu et al., 2025): we still generate $n$ rollouts, but backpropagate through only $m < n$ rollouts chosen to maximize reward variance. Intuitively, this retains the most contrastive rollouts (high vs. low reward), which carry the strongest learning signal, while reducing policy-update computation by roughly a factor of $n/m$.

Formally, for prompt $s_{i,k}$ with sorted rewards $\{r_{i,k+1,1} \leq r_{i,k+1,2} \leq \cdots \leq r_{i,k+1,n}\}$, the optimal subset selects the

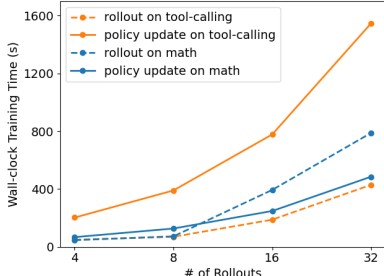

*Figure 6.* Wall-clock training time breakdown for Qwen3-4B on multi-turn tool-calling and math reasoning tasks. Policy update time (orange) dominates rollout generation (blue) and grows rapidly with the number of rollouts, revealing a severe computational asymmetry that intensifies with rollout count.

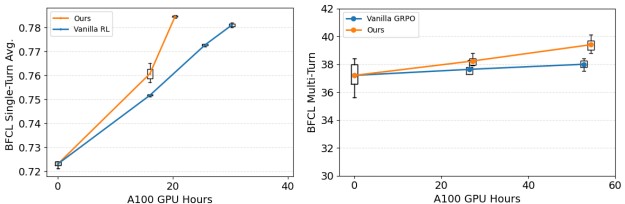

*Figure 7.* Comparison of vanilla GRPO and GRPO with efficiency methods for single-turn (left) and multi-turn (right) setting. Given the same or less wall-clock time, our method yields strictly better performance, demonstrating more effective utilization of rollout and policy update computation.

extremes:

$$\mathcal{S}^* = \{1, \dots, m'\} \cup \{n - (m - m') + 1, \dots, n\}, \quad (7)$$

i.e., $m'$ lowest-reward and $m - m'$ highest-reward rollouts. When rewards are binary and $m$ is even, this reduces to selecting $m/2$ rollouts from each reward class whenever both are present.

### 4.3. Results and Discussion

**Efficiency at matched wall-clock time.** Fig. 7 compares vanilla GRPO with GRPO augmented by our two efficiency techniques. Across both settings (Qwen2.5-3B-Instruct single-turn and Qwen3-4B multi-turn), our method achieves higher accuracy under the same wall-clock budget, indicating more effective use of rollout and update computation. Measured by the time required to reach comparable performance, this corresponds to a $1.7\times$ and $2.6\times$ speedup in single-turn and multi-turn settings, respectively. In the multi-turn setting, both methods reach similar performance early on (around 27 GPU-hours), but the efficiency-augmented variant continues to improve with further training, consistent with reducing redundant rollouts and amortizing expensive policy updates over more informative samples.

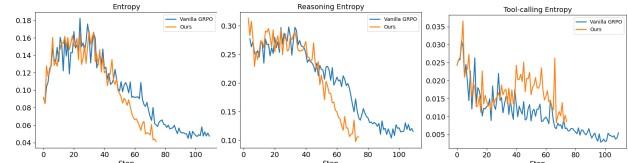

*Figure 8.* Entropy dynamics during tool-calling RL training. From left to right: total response token entropy, thinking-token entropy, and tool-calling token entropy, plotted as a function of training steps.

**Comparison to other models.** We report a broader comparison against other representative open-source and closed-source models in Table 4. Baseline results are extracted from the official BFCL benchmark under the same evaluation version. Qwen3-4B with a stronger system prompt already demonstrates competitive performance. After applying our training method, the resulting model further improves multi-turn tool-calling performance, achieving an average score of 39.4.

**Generalization beyond BFCL.** On ACEBench [2] as shown in Table 3, our RL-trained model improves substantially over the base model under the benchmark default prompt (+12.1 overall accuracy points), outperforming Nova1-Lite and several open-source baselines. This suggests that the gains are not limited to BFCL-specific formatting, but transfer to a distinct evaluation suite with different tool-use patterns.

**Entropy dynamics.** Model responses contain both free-form reasoning and structured tool-calling content. In Fig. 8, we track average token entropy for the full response and separately for the reasoning and tool-calling segments, comparing vanilla GRPO against GRPO with our efficiency methods (Qwen2.5-3B-Instruct, single-turn setting). We observe a common pattern: total and reasoning entropy rise slightly early in training and then decrease, consistent with RL driving the policy toward more confident outputs. With our efficiency methods, this entropy reduction happens earlier, suggesting faster convergence for the reasoning component. In contrast, tool-calling entropy shows a transient spike, indicating increased exploration in tool invocation before stabilizing. Overall, reasoning entropy remains substantially higher than tool-calling entropy, reflecting the more constrained and deterministic structure of tool outputs.

**Downstream task evaluation.** To check whether tool-calling RL impacts general capabilities, we evaluate the base and RL-trained models on HellaSwag (Zellers et al., 2019),

---

[2]Our initial RL-trained model achieves 66.0% average accuracy. We then expanded the sampled training data from 2.6k to a 6k subset and perform training from scratch using the same pipeline.

*Table 3.* Tool-calling performance on ACEBench for English data on all categories. The best score across each category is displayed in bold.

| | Agent | | Normal | | | | | Special | | | Overall |
|---|---|---|---|---|---|---|---|---|---|---|---|
| | Multistep | Multiturn | Atom | Multiturn | Singleturn | Preference | Sim. API | Error | Incom. | Irrelevant | |
| *Closed-source models* | | | | | | | | | | | |
| Claude-4-Sonnet | 15.0 | 53.3 | **94.0** | 79.0 | 86.5 | **76.0** | **86.0** | **96.0** | 76.0 | **94.0** | **81.8** |
| Nova-1-Pro | 20.0 | 56.7 | 94.3 | **81.5** | **88.5** | 74.0 | 76.0 | 62.0 | 44.0 | 84.0 | 81.4 |
| Nova-1-Lite | 20.0 | **60.0** | 86.0 | 60.0 | 78.5 | 68.0 | 74.0 | 14.0 | 46.0 | 78.0 | 73.4 |
| *Open-source models* | | | | | | | | | | | |
| Gemma3-27B-Instruct | 20.0 | 56.7 | 88.3 | 55.0 | 81.0 | 66.0 | 76.0 | 70.0 | 32.0 | 90.0 | 76.8 |
| Magistral-Small-24B-2509 | 20.0 | 50.0 | 91.0 | 58.0 | 82.0 | 70.0 | 74.0 | 40.0 | 18.0 | 48.0 | 67.4 |
| Qwen3-4B (base) | 10.0 | 20.0 | 77.3 | 64.7 | 66.5 | 64.0 | 76.0 | 58.0 | 84.0 | 74.0 | 65.4 |
| Qwen3-4B-RL (ours) | **25.0** | 30.0 | 90.0 | 79.5 | 80.0 | 70.0 | 80.0 | 72.0 | **90.0** | 92.0 | 77.5 |

*Table 4.* Tool-calling performance comparison on BFCL on multi-turn and single-turn categories with other models.

| Models | Multi-turn | Single-turn | Avg. |
|---|---|---|---|
| *Closed-source models* | | | |
| Claude Sonnet 4.5 (FC) | 61.4 | 84.9 | 73.2 |
| Gemini-3-Pro-Preview (FC) | 63.1 | 83.8 | 73.4 |
| GPT-4.1-2025-04-14 (FC) | 38.9 | 76.4 | 57.7 |
| Grok-4-0709 (FC) | 33.9 | 80.5 | 57.2 |
| Nova-2-Lite-v1:0 (FC) | 2.1 | 83.9 | 43.0 |
| *Open-source models* | | | |
| Llama-4-Scout-17B-16E -Instruct (FC) | 9.0 | 82.1 | 45.5 |
| Mistral-Large-2411 (FC) | 14.1 | 83.3 | 48.7 |
| Qwen3-235B-A22B-Instruct -2507 (FC) | 45.4 | 53.2 | 49.3 |
| Qwen3-4B w. bfcl | 22.7±0.9 | 83.9±0.5 | 53.3±0.5 |
| Qwen3-4B w. stronger | 37.2±1.4 | 84.8±0.7 | 61.0±0.8 |
| Qwen3-4B-RL (ours) | 39.4±0.7 | 84.8±0.9 | 62.1±0.5 |

*Table 5.* Downstream task evaluation results for base model (Qwen3-4B) and model after RL (Qwen3-4B-RL).

| | HellaSwag | MMLU | TruthfulQA | Winogrande |
|---|---|---|---|---|
| Qwen3-4B (base) | 52.1 | 68.3 | 36.7 | 65.8 |
| Qwen3-4B-RL (ours) | 52.3 | 68.3 | 37.0 | 66.5 |

MMLU (Hendrycks et al., 2020), TruthfulQA (Lin et al., 2022), and WinoGrande (Sakaguchi et al., 2021) using the official `lm-evaluation-harness`[3] (Gao et al., 2024). As shown in Table 5, we verify that tool-calling RL does not meaningfully degrade downstream performance relative to the base model.

## 5. Related Work

**Tool learning for LLMs.** Equipping LLMs with external tools is a practical and research-critical direction for extending capabilities beyond parametric knowledge (Schick et al., 2023; Yao et al., 2022). Tools commonly include

web search (Vu et al., 2024; Jin et al., 2025), code execution (Chen et al., 2022; Gao et al., 2023), and domain-specific APIs. Existing approaches broadly fall into two lines. Early work elicits tool use via prompting and orchestration without additional training (Yao et al., 2022; Shen et al., 2023; Paranjape et al., 2023). Subsequent work improves reliability through post-training, including supervised fine-tuning (SFT) on tool-use traces (Schick et al., 2023; Qin et al., 2023; Patil et al., 2024; Zhang et al., 2025a) and, more recently, reinforcement learning (RL) for tool-calling and agent behavior (Qian et al., 2025; Zhang et al., 2025c). Our work complements this literature by studying *how* tool-calling progress is measured and *how* RL tool-calling can be made substantially more efficient, rather than proposing a new tool-use dataset or a new agent architecture.

**Evaluation sensitivity and reproducibility.** Across machine learning, apparent gains can be fragile artifacts of evaluation pipelines, including implementation details, prompting, and reporting conventions, rather than robust improvements (Dehghani et al., 2021; Liao et al., 2021; Islam et al., 2017; Hochlehnert et al., 2025). This concern is especially acute in RL, where results are often sensitive to seemingly minor choices such as random seeds and environment or training details (Henderson et al., 2018; Agarwal et al., 2021; Chan et al., 2019; Patterson et al., 2024).

**RL efficiency.** A line of work has aimed to improve RL efficiency by addressing the issue of zero-variance prompts. Dynamic sampling (Yu et al., 2025) over-samples and filters out zero-variance prompts at each training step, yet we observe such sampling actually significantly increases training time. Methods like (Xiong et al., 2025) proposes to allocate rollout budget among different prompts, which reduces the number of required training steps but increases the per-step training time. Le et al. (2025) further proposes to utilize zero-variance prompts by optimizing on their entropy.

---

[3]https://github.com/EleutherAI/lm-evaluation-harness

## 6. Conclusions

We presented a comprehensive study on the effectiveness and efficiency of tool-calling evaluation and RL training for LLMs. On the effectiveness side, we conducted the first systematic analysis of factors affecting tool-calling benchmark reliability. Our findings reveal that seemingly minor design choices often neglected by the community, including those beyond benchmark default settings, can lead to substantial performance differences. These results call for more rigorous evaluation protocols in the tool-calling community. On the training efficiency side, we identify two major computational bottlenecks in RL-based tool-calling training: the prevalence of zero-variance prompts and the high cost of policy updates. We proposed two complementary techniques that together achieve much faster speedup without degrading performance.

## Impact Statement

This paper studies the effectiveness and efficiency of agentic tool-calling evaluation and RL training. By improving evaluation transparency and reducing unnecessary computation, our methods can support more reliable comparisons and lower the cost and environmental footprint of developing tool-using agents. At the same time, more efficient training may lower the barrier to building and deploying capable agents, which could accelerate both beneficial applications (e.g., automation and decision support) and potential misuse (e.g., scalable manipulation or abuse of external tools). We do not introduce new tools, datasets containing sensitive information, or capabilities explicitly aimed at bypassing safeguards; our contributions focus on evaluation methodology and training efficiency. We encourage future work to pair improved tool-calling capability with rigorous safety evaluation, logging/monitoring, and responsible deployment practices.

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

D. Reproducibility of benchmarked deep reinforcement
learning tasks for continuous control. *arXiv preprint
arXiv:1708.04133*, 2017.

Jin, B., Zeng, H., Yue, Z., Yoon, J., Arik, S., Wang, D.,
Zamani, H., and Han, J. Search-r1: Training llms to
reason and leverage search engines with reinforcement
learning. *arXiv preprint arXiv:2503.09516*, 2025.

Le, T.-L. V., Jeon, M., Vu, K., Lai, V., and Yang, E. No
prompt left behind: Exploiting zero-variance prompts in
llm reinforcement learning via entropy-guided advantage
shaping. *arXiv preprint arXiv:2509.21880*, 2025.

Liao, T., Taori, R., Raji, I. D., and Schmidt, L. Are we
learning yet? a meta review of evaluation failures across
machine learning. In *Thirty-fifth Conference on Neural In-
formation Processing Systems Datasets and Benchmarks
Track (Round 2)*, 2021.

Lin, S., Hilton, J., and Evans, O. Truthfulqa: Measuring
how models mimic human falsehoods. In *Proceedings of

the 60th annual meeting of the association for computa-
tional linguistics (volume 1: long papers)*, pp. 3214–3252,
2022.

Liu, W., Huang, X., Zeng, X., Hao, X., Yu, S., Li, D.,
Wang, S., Gan, W., Liu, Z., Yu, Y., et al. Toolace: Win-
ning the points of llm function calling. *arXiv preprint
arXiv:2409.00920*, 2024.

Meta. llama4 system card. https://ai.meta.com/
blog/llama-4-multimodal-intelligence/,
2025. Accessed: 2025-4.

MistralAI. Mistral small models. https://huggingf
ace.co/mistralai/Magistral-Small-2509,
2025.

Openai. Gpt5 system card. https://cdn.openai.c
om/gpt-5-system-card.pdf, 2025. Accessed:
2025-08.

Paranjape, B., Lundberg, S., Singh, S., Hajishirzi, H., Zettle-
moyer, L., and Ribeiro, M. T. Art: Automatic multi-step
reasoning and tool-use for large language models. *arXiv
preprint arXiv:2303.09014*, 2023.

Patil, S. G., Zhang, T., Wang, X., and Gonzalez, J. E. Go-
rilla: Large language model connected with massive apis.
*Advances in Neural Information Processing Systems*, 37:
126544–126565, 2024.

Patil, S. G., Mao, H., Yan, F., Ji, C. C.-J., Suresh, V., Stoica,
I., and Gonzalez, J. E. The berkeley function calling
leaderboard (bfcl): From tool use to agentic evaluation
of large language models. In *Forty-second International
Conference on Machine Learning*, 2025.

Patterson, A., Neumann, S., White, M., and White, A. Em-
pirical design in reinforcement learning. *Journal of Ma-
chine Learning Research*, 25(318):1–63, 2024.

Prabhakar, A., Liu, Z., Zhu, M., Zhang, J., Awalgaonkar,
T., Wang, S., Liu, Z., Chen, H., Hoang, T., Niebles, J. C.,
et al. Apigen-mt: Agentic pipeline for multi-turn data
generation via simulated agent-human interplay. *arXiv
preprint arXiv:2504.03601*, 2025.

Qian, C., Acikgoz, E. C., He, Q., Wang, H., Chen, X.,
Hakkani-Tür, D., Tur, G., and Ji, H. Toolrl: Reward is
all tool learning needs. *CoRR*, abs/2504.13958, 2025.
doi: 10.48550/ARXIV.2504.13958. URL https:
//doi.org/10.48550/arXiv.2504.13958.

Qin, Y., Liang, S., Ye, Y., Zhu, K., Yan, L., Lu, Y., Lin, Y.,
Cong, X., Tang, X., Qian, B., et al. Toolllm: Facilitating
large language models to master 16000+ real-world apis.
*arXiv preprint arXiv:2307.16789*, 2023.

Reimers, N. and Gurevych, I. Sentence-bert: Sentence embeddings using siamese bert-networks. In *Proceedings of the 2019 conference on empirical methods in natural language processing and the 9th international joint conference on natural language processing (EMNLP-IJCNLP)*, pp. 3982–3992, 2019.

Reuel, A., Hardy, A., Smith, C., Lamparth, M., Hardy, M., and Kochenderfer, M. J. Betterbench: Assessing ai benchmarks, uncovering issues, and establishing best practices. *Advances in Neural Information Processing Systems*, 37:21763–21813, 2024.

Sakaguchi, K., Bras, R. L., Bhagavatula, C., and Choi, Y. Winogrande: An adversarial winograd schema challenge at scale. *Communications of the ACM*, 64(9):99–106, 2021.

Schick, T., Dwivedi-Yu, J., Dessì, R., Raileanu, R., Lomeli, M., Hambro, E., Zettlemoyer, L., Cancedda, N., and Scialom, T. Toolformer: Language models can teach themselves to use tools. *Advances in Neural Information Processing Systems*, 36:68539–68551, 2023.

Schulman, J., Wolski, F., Dhariwal, P., Radford, A., and Klimov, O. Proximal policy optimization algorithms. *arXiv preprint arXiv:1707.06347*, 2017.

Shao, Z., Wang, P., Zhu, Q., Xu, R., Song, J., Bi, X., Zhang, H., Zhang, M., Li, Y., Wu, Y., et al. Deepseekmath: Pushing the limits of mathematical reasoning in open language models. *arXiv preprint arXiv:2402.03300*, 2024.

Shen, Y., Song, K., Tan, X., Li, D., Lu, W., and Zhuang, Y. Hugginggpt: Solving ai tasks with chatgpt and its friends in hugging face. *Advances in Neural Information Processing Systems*, 36:38154–38180, 2023.

Sheng, G., Zhang, C., Ye, Z., Wu, X., Zhang, W., Zhang, R., Peng, Y., Lin, H., and Wu, C. Hybridflow: A flexible and efficient rlhf framework. *arXiv preprint arXiv:2409.19256*, 2024.

Shi, Q., Zytek, A., Razavi, P., Narasimhan, K., and Barres, V. $\tau$-knowledge: Evaluating conversational agents over unstructured knowledge. *arXiv preprint arXiv:2603.04370*, 2026.

Team, G., Kamath, A., Ferret, J., Pathak, S., Vieillard, N., Merhej, R., Perrin, S., Matejovicova, T., Ramé, A., Rivière, M., et al. Gemma 3 technical report. *arXiv preprint arXiv:2503.19786*, 2025.

Vu, T., Iyyer, M., Wang, X., Constant, N., Wei, J., Wei, J., Tar, C., Sung, Y.-H., Zhou, D., Le, Q., et al. Freshllms: Refreshing large language models with search engine augmentation. In *Findings of the Association for Computational Linguistics: ACL 2024*, pp. 13697–13720, 2024.

xAI. grok4.1 fast. `https://x.ai/news/grok-4-1-fast`, 2025. Accessed: 2025-11.

Xiong, W., Ye, C., Liao, B., Dong, H., Xu, X., Monz, C., Bian, J., Jiang, N., and Zhang, T. Reinforce-ada: An adaptive sampling framework for reinforce-style llm training. *arXiv e-prints*, pp. arXiv–2510, 2025.

Xu, Y. E., Savani, Y., Fang, F., and Kolter, J. Z. Not all rollouts are useful: Down-sampling rollouts in llm reinforcement learning. *arXiv preprint arXiv:2504.13818*, 2025.

Yang, A., Li, A., Yang, B., Zhang, B., Hui, B., Zheng, B., Yu, B., Gao, C., Huang, C., Lv, C., et al. Qwen3 technical report. *arXiv preprint arXiv:2505.09388*, 2025.

Yao, S., Zhao, J., Yu, D., Du, N., Shafran, I., Narasimhan, K. R., and Cao, Y. React: Synergizing reasoning and acting in language models. In *The eleventh international conference on learning representations*, 2022.

Yu, Q., Zhang, Z., Zhu, R., Yuan, Y., Zuo, X., Yue, Y., Dai, W., Fan, T., Liu, G., Liu, L., et al. Dapo: An open-source llm reinforcement learning system at scale. *arXiv preprint arXiv:2503.14476*, 2025.

Zellers, R., Holtzman, A., Bisk, Y., Farhadi, A., and Choi, Y. Hellaswag: Can a machine really finish your sentence? *arXiv preprint arXiv:1905.07830*, 2019.

Zhang, J., Lan, T., Zhu, M., Liu, Z., Hoang, T. Q., Kokane, S., Yao, W., Tan, J., Prabhakar, A., Chen, H., et al. xlam: A family of large action models to empower ai agent systems. In *Proceedings of the 2025 Conference of the Nations of the Americas Chapter of the Association for Computational Linguistics: Human Language Technologies (Volume 1: Long Papers)*, pp. 11583–11597, 2025a.

Zhang, K., Jiao, W., Du, K., Lu, Y., Liu, W., Zhang, W., Zhang, L., and Yu, Y. Looptool: Closing the data-training loop for robust llm tool calls, 2025b.

Zhang, S., Dong, Y., Zhang, J., Kautz, J., Catanzaro, B., Tao, A., Wu, Q., Yu, Z., and Liu, G. Nemotron-research-tool-n1: Exploring tool-using language models with reinforced reasoning. *CoRR*, abs/2505.00024, 2025c. doi: 10.48550/ARXIV.2505.00024.

Zheng, H., Zhou, Y., Bartoldson, B. R., Kailkhura, B., Lai, F., Zhao, J., and Chen, B. Act only when it pays: Efficient reinforcement learning for llm reasoning via selective rollouts. *arXiv preprint arXiv:2506.02177*, 2025.

In this Appendix, we present the following:

- Employed System Prompts;

- Similarity Between Training and Test Data;

- Models for Evaluation and Training;

- Data Preprocessing and Filtering;

- Experimental Setup in Section 3.5;

- Hyperparameters;

- Other Results

## A. Employed System Prompts

We present the default system prompt in BFCL and another slightly modified system prompt in Fig. 9 and 10, respectively.

To disentangle system prompt length from added instructions, we construct two intermediate variants: (1) duplicating the default prompt once to match length, and (2) a **partially strengthened** prompt with only instructions 1&2. Since multi-turn variance can reach up to 3% (Takeaway 3.1), we report one inference run in Table 6.

*Table 6.* Influence of averaged input token length and added instructions. BFCL multi-turn for Qwen3-4B with default BFCL prompt, copying prompt, stronger one with 1&2 and full instructions.

|  | default | copying default | stronger (1&2) | stronger (full) |
|---|---|---|---|---|
| Avg. input token length | 228 | 452 | 302 | 448 |
| BFCL multi-turn | 22.9 | 23.6 | 36.0 | 37.5 |

Overall, simply increasing input prompt length does not have a positive effect, whereas adding task-relevant instructions yields substantial gains (22.9 $\rightarrow$ 36.0/37.5), suggesting that the improvement comes from prompt content rather than length alone.

## B. Similarity Between Training and Test Data

We use sentence-BERT embeddings (Reimers & Gurevych, 2019) to measure cosine similarity between each training dataset in Section 3.5/Section 4 and the BFCL multi-turn base category.

*Table 7.* Similarity analysis between training data and BFCL benchmark.

|  | cross (train↔BFCL) | within-train | within-BFCL |
|---|---|---|---|
| multi-turn (§3.5) | 0.063 | 0.083 | 0.132 |
| single-turn (§3.5) | 0.065 | 0.100 | 0.132 |
| multi-turn (§4) | 0.082 | 0.126 | 0.132 |

We observe that the cross-domain similarity between the §3.5 training data (multi-turn/single-turn) and BFCL multi-turn is substantially lower than the within-BFCL similarity (0.132) and within-training similarity (0.083, 0.100). This is consistent with the empirical results in Table 2: training on such data might not help BFCL multi-turn tasks. In comparison, the customized multi-turn data in §4 exhibits higher similarity to BFCL, and correspondingly leads to improved performance.

## C. Models for Evaluation and Training

In Section 3, we run five commonly used models, Qwen3-4B [4], Qwen3-8B [5], Qwen2.5-7B-Instruct [6], Llama3.1-8B-Instruct [7] and Llama3.2-3B-Instruct [8]. Our training primarily focuses on Qwen3-4B for multi-turn setting and Qwen2.5-3B-Instruct [9] for single-turn setting. We compare with baselines of open-source models Gemma3-Instruct 27B (Team et al., 2025) [10], Magistral-Small-2509 24B (MistralAI, 2025)) [11] on ACEBench.

## D. Data Preprocessing and Filtering

We describe the data preprocessing and filtering procedures used in Section 2.3. For single-turn data, we perform 2-rollout filtering using a policy model trained on the original N1 dataset for 80 steps. This model is capable of producing both reasoning traces and tool-calling tags. Prompts for which both rollouts produce either correct or incorrect answers are removed. For multi-turn data, we first extract prompts with a number of turns between 2 and 6, resulting in a 6k subset. We then apply 8-rollout filtering using a stronger model, and discard any prompt where the model fails to produce a correct answer in all rollouts. Finally, we exclude prompts that involve more than one output tool call, resulting a 2.6k subset.

## E. Experimental Setup in Section 3.5

We conduct a controlled experiment that isolates training format by fine-tuning separate models on *pure* single-turn

---

[4] https://huggingface.co/Qwen/Qwen3-4B

[5] https://huggingface.co/Qwen/Qwen3-8B

[6] https://huggingface.co/Qwen/Qwen2.5-7B-Instruct

[7] https://huggingface.co/meta-llama/Llama-3.1-8B-Instruct

[8] https://huggingface.co/meta-llama/Llama-3.2-3B-Instruct

[9] https://huggingface.co/Qwen/Qwen2.5-3B-Instruct

[10] https://huggingface.co/google/gemma-3-27b-it

[11] https://huggingface.co/mistralai/Magistral-Small-2509

```
You are an expert in composing functions.You are given a question and a set of possible functions. Based on the
question, you will need to make one or more function/tool calls to achieve the purpose. If none of the functions can
 be used, point it out. If the given question lacks the parameters required by the function, also point it out.

You should only return the function calls in your response.

If you decide to invoke any of the function(s), you MUST put it in the format of [func_name1(params_name1=
params_value1, params_name2=params_value2...), func_name2(params)]. You SHOULD NOT include any other text in the
response.

At each turn, you should try your best to complete the tasks requested by the user within the current turn. Continue
 to output functions to call until you have fulfilled the user's request to the best of your ability. Once you have
no more functions to call, the system will consider the current turn complete and proceed to the next turn or task.

Here is a list of functions in json format that you can invoke.
[
...
]
```

*Figure 9.* Default BFCL evaluation system prompt.

vs. *pure* multi-turn data. We construct two matched training datasets derived from xLAM (Zhang et al., 2025a) and ToolACE (Liu et al., 2024), following the preprocessing method in Zhang et al. (2025c). (i) **Multi-turn set:** we use the released multi-turn trajectories directly. (ii) **Single-turn set:** to avoid overly trivial instances, we filter examples by sampling 8 rollouts from the base policy and discarding prompts where the model succeeds on all rollouts ; we then subsample single-turn data to match the multi-turn set size (about 0.7k examples each). We RL fine-tune two Qwen3-4B models for 20 epochs with identical hyperparameters, and evaluate on both BFCL single-turn and multi-turn datasets

Reward design: Given trajectory prefix $s_{i,k}$, for single-turn setting, we follow Zhang et al. (2025c) to encourage the small model to explicitly output thinking process by designing a reward that jointly contains both format correctness (i.e., containing both <think>, </think>, and <tool_call>, </tool_call >tags) and tool call correctness (output tools exactly math ground truth tools $T_{i,k+1} = T^*_{i,k+1}$):

$$r(s_{i,k}) = \begin{cases} 1, & \text{if FormatCorrect} \wedge \text{ToolCallMatch} \\ 0, & \text{otherwise} \end{cases}$$
(8)

For multi-turn setting, we remove the limit of format correctness and only judge the output based on tool call correctness.

## F. Hyperparameters

We use a learning rate of 1e-6 and remove the KL and entropy terms. For multi-turn training, we use a batch size of 256 for 4 epochs with temperature 1.0 to encourage exploration, and set the maximum prompt length to 4096. We also train on another 6k subset with maximum prompt length to

2048 with truncation and find that this setting significantly improves performance on ACEBench. We guess the reason could be setting a shorter maximum prompt length might provide an implicit regularization. For single-turn training, we use a batch size of 128 for 10 epochs with temperature 0.7 following Zhang et al. (2025c), and set the maximum prompt length to 2048. All multi-turn experiments are performed on 8 A100 GPUs. We adopt the clip-higher trick and use the clipping range of $[0.2, 0.28]$ following setting on math tasks (Yu et al., 2025). We fix the number of rollouts to be 8 in all settings, except for multi-turn with efficient GRPO, where we use 16 rollouts. For our method, we tune $k = \{1, 2\}$ for pre-rollout filtering and $m = \{4, 8/16\}$ for max-variance down-sampling in both setting. We find in single-turn, max-variance down-sampling (Xu et al., 2025) does not work very well and can even hurt the performance.

## G. Other Results

To evaluate whether the observed issues of wasted computation in Section 4 are model-specific, we extend our experiments to larger-scale models, including Llama3.1-8B, Qwen2.5-7B and Qwen3-8B. First, as shown in Fig. 11, we observe the same pattern of a high zero-variance prompt ratio in tool-calling RL. This issue becomes more pronounced as model size increases. Second, Fig. 11 shows the policy update phase continues to dominate the wall-clock training time for the 8B model. Overall, this suggests that the observed phenomena are not a consequence of limited model capacity, but a more general characteristic.

---

> **Better BFCL Evaluation System Prompt**
>
> ```
> You are an expert in composing functions. You are given a question and a set of possible functions. Based on the
> question, you will need to make one or more function/tool calls to achieve the purpose. If none of the functions can
>  be used, point it out. If the given question lacks the parameters required by the function, also point it out. If
> the result of tool calls has fulfilled the user's request, summary the answer.
>
> **Important Notes**
> 1. If a tool call completes the user's request, return a concise plain-text summary and stop. Do not call any
> additional tools. If no tool is suitable, state that explicitly. If the user's input lacks required parameters, ask
> for clarification.
> 2. During each tool invocation, it is important to carefully examine the corresponding tool's description and
> constraints. Ensure that the required fields of the tool are strictly satisfied and that parameter types conform to
> the definitions. If function calls uses the default parameter value, it is not necessary to specify the value during
>  the call.
> 3. For complex requests requiring multiple function calls or where parameters of subsequent function calls depend on
>  depend on the results of previous calls, decompose the task into. You only need to return the result of the first
> step. Please never use fictitious parameters or placeholders.
> 4. When encountering errors in multi-turn conversations, analyze the tool feedback to understand what went wrong, e.
> g., whether it's incorrect tool selection, invalid parameters, or missing steps. Then retry with corrected tool
> calls, iterating until the task is successfully completed or determined to be unachievable with available tools.
> Once you have no more functions to call, the system will consider the current turn complete and proceed to the next
> turn or task.
>
> The current time is Unknown.
>
> # Tools
>
> You may call one or more functions to assist with the user query. You are provided with function signatures within <
> tools></tools> XML tags:
> <tools>
> [
> ...
> ]
> </tools>
>
> For each function call, return a json object with function name and arguments within <tool_call></tool_call> XML
> tags:
> <tool_call>
> {"name": <function-name>, "arguments": <args-json-object>}
> </tool_call>
> ```

*Figure 10.* A stonger BFCL evaluation system prompt by slightly manual modification.

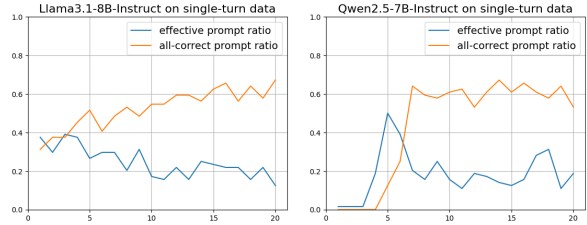

*Figure 11.* Ratio of zero-variance vs. non-zero-variance prompts during RL training of Llama3.1-8B-Instruct and Qwen2.5-7B-Instruct on the single-turn tool-calling dataset on the first 20 steps. The batch size here is half of that in Fig. 4, which even stretches the x-axis compared to small size. Blue: ratio of prompts whose rollout rewards exhibit variance (i.e., useful learning signal). Orange: ratio of prompts with all rollouts achieving the maximum reward. In both cases, only around 20% prompts are effective prompts, indicating significant rollout waste.

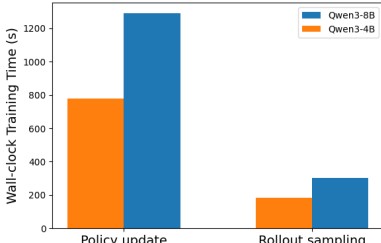

*Figure 12.* Wall-clock training time breakdown for Qwen3-4B and Qwen3-8B on multi-turn tool-calling. The high computational cost during policy updates persists as model size increases.

