# OpenReview forum: "On Effectiveness and Efficiency of Agentic Tool-calling and RL Training"
_ICML.cc/2026/Conference — ICML 2026 regular_

### Official Review · Reviewer_PoMF · 2026-02-26

**Soundness:** 3
**Presentation:** 3
**Significance:** 3
**Originality:** 3
**Overall Recommendation:** 4
**Confidence:** 4

**Summary:**

This paper systematically investigates two critical dimensions of LLM tool-calling: effectiveness in evaluation and efficiency in training. By analyzing with BFCL, the authors demonstrate that reported performance is highly sensitive to implementation details, such as random seeds, chat templates, and system prompts, with small changes in system prompts yielding siginificant gains. For training inefficiencies, the authors identify that up to 80% of RL prompts provide no learning signal (zero-variance) and that policy updates are disproportionately expensive due to long sequence lengths. They introduce two primary optimizations: online pre-rollout filtering, which skips redundant prompts that the model has already mastered, and variance-aware rollout down-sampling, which reduces optimization costs by updating only on the most informative samples. Together, these methods significantly accelerate training wall-clock time while maintaining or improving performance across benchmarks like BFCL and ACEBench.

**Compliance With Llm Reviewing Policy:**

Affirmed.

**Final Justification:**

The overall analysis presented in the paper is interesting, and several of my concerns have been partially addressed during the rebuttal. However, a key issue that prevents the paper from being fully convincing remains unresolved: the experimental evaluation is conducted on only a single benchmark. Therefore, I maintain my original score.

**Key Questions For Authors:**

1. Qwen2.5-7B is not a reasoning model. How are the results Qwen2.5-7B in Figure3-middle derived?
2. Why do you choose  Qwen2.5-3B-Instruct for single-turn tool-calling and Qwen3-4B for multi-turn tool-calling? Have you tried other models?

**Limitations:**

The limitation mainly lies in the experimental scale. More benchmark results with diverse sizes of models should improve the persuasiveness.

**Strengths And Weaknesses:**

Strengths:
1. The paper conducts interesting analysis on evaluation robustness and learning efficiency, which should be useful for the field.
2. Systematic analysis on evaluation robustness reveals several useful insights, leading potential design for future benchmarks and model evaluation.
3. Efficiency gain is obvious with the two methods introduced by the paper, according to the experiments.

Weaknesses:
1. The evaluation analysis is limited to the BFCL benchmark, making it difficult to assess whether it is just a prolem of one specific benchmark.
2. The online pre-rollout filtering trick seems like a common practice in GRPO training.
3. It lack comparisons to vanilla RL or with other RL tricks in final performance, for example, in Table 3.
4. The results in Table 2 are not convincing. It is more likely because the domain mismatch between the training data and testset. The finding "Multi-turn training does not automatically yield stronger multi-turn tool-calling" is problematic. In-domain training improves performance should be common understanding.

---

> ### Author Rebuttal · Authors · 2026-03-31
>
> The authors thank the reviewer for recognizing the evaluation analysis being interesting and useful and constructive feedback! We provide detailed replies below:
>
> **Q1**: The evaluation analysis is limited to the BFCL benchmark, making it difficult to assess whether it is just a problem of one specific benchmark.
> **A1**: Thank you for this question. In this paper, we focus on general tool-use capabilities and therefore choose BFCL as a representative case study. It’s still widely used by many papers and technical reports. BFCL covers a diverse set of domains and function-calling scenarios including coding, terminal, web, and tool interactions. As such, we expect our findings, particularly those related to evaluation sensitivity and training efficiency, to **generalize** to other agent benchmarks (e.g., other web and coding).
> Some evidence from existing papers: In an Android coding benchmark paper [1], authors perform seed-variance evaluation and find “Agent performance **varies** much more using different **seeds**.”
> [1]. AndroidWorld: A Dynamic Benchmarking Environment for Autonomous Agents, ICLR 2025
>
> **Q2**: The online pre-rollout filtering trick seems like a common practice in GRPO training.
> **A2**: Thank you for the comment. While filtering uninformative prompts may appear intuitive, to the best of our knowledge, online pre-rollout filtering has not been systematically studied for agentic RL.
>
> **Q3**: Qwen2.5-7B is not a reasoning model. How are the results Qwen2.5-7B in Figure3-middle derived?
> **A3**: Thank you for pointing this out. We agree that Qwen2.5-7B is not a reasoning model in the same sense as Qwen3. In evaluation, sometimes given the step-by-step instruction prompt, it can produce some reasoning-like text in some cases, which is why it was initially included in this analysis. However, we agree that this may still be misleading. We will update the paper to remove this model from the plot.
>
> **Q4**: Why do you choose Qwen2.5-3B-Instruct for single-turn tool-calling and Qwen3-4B for multi-turn tool-calling? Have you tried other models?
> **A4**: We chose Qwen2.5-3B-Instruct for single-turn setting and Qwen3-4B for multi-turn setting as representative models due to their strong tool-calling performance in the setups.
> Previously we also experimented with Llama 3.2 in the single-turn setting and observed that it can also improve after RL. However, its overall performance is weaker than the corresponding Qwen2.5 model in our experiments (see link [here](https://anonymous.4open.science/r/Anonymity-5EF5/README.md). For multi-turn, Llama 3.2 has very weak capability (see Fig. 1 in the paper) and almost cannot improve with RL in our setting.
> Therefore, we use two Qwen models as the main representative model in the paper.

---

> > ### Author Rebuttal · Reviewer_PoMF · 2026-04-02
> >
> > I am not satisfied with the answers to Q1. While BFCL is a representative benchmark, the corresponding analysis results still cannot be ensured that "whether it is just a problem of one specific benchmark". Also, my weaknesses 3 and 4 have not been addressed.

---

> > > ### Author Response · Authors · 2026-04-07
> > >
> > > We thank the reviewer for the further comments. We provide detailed replies below:
> > >
> > > **Q1**: The results in Table 2 are not convincing. It is more likely because the domain mismatch between the training data and testset. The finding "Multi-turn training does not automatically yield stronger multi-turn tool-calling" is problematic. In-domain training improves performance should be common understanding.
> > > **A1**: We appreciate the reviewer's concern. First, we’d like to clarify that our claim from Table 2 in the paper is **not** that “multi-turn training does not improve multi-turn tool-calling” in general. Rather, our finding is: **current** multi-turn trajectories can be a noisy training signal and trajectory **quality** matters more than simply increasing the number of multi-turn prompts.
> > > Regarding domain mismatch, we **agree** with the reviewer and measure cosine similarity (using sentence-BERT embeddings) between each training dataset and the BFCL multi-turn base category.
> > >
> > > | | cross (train↔BFCL) | within-train | within-BFCL |
> > > |-------|-------|-----|------|
> > > | multi-turn (§3.5) | 0.063 | 0.083 | 0.132 |
> > > | single-turn (§3.5) | 0.065 | 0.100 | 0.132 |
> > > | multi-turn (§4) | 0.082 | 0.126 | 0.132 |
> > >
> > > Table 1. Similarity analysis between training data and BFCL benchmark.
> > >
> > > We observe that the **cross-domain similarity** between the §3.5 training data (multi-turn/single-turn in) and BFCL is substantially lower than the within-BFCL similarity (0.132) and within-training similarity (0.083, 0.100).
> > > This finding is consistent with the empirical results in Table 2: training on such data degrades BFCL multi-turn tasks. In comparison, the customized multi-turn data in §4 exhibits higher similarity to BFCL, and correspondingly leads to improved performance.
> > > We thank the reviewer again and will include such alignment analysis in the final version of the paper.
> > >
> > > **Q2**: It lacks comparisons to vanilla RL or with other RL tricks in final performance
> > > **A2**: We include comparisons with vanilla RL and report the final performance here.
> > >
> > > | | BFCL multi-turn (avg.) | BFCL single-turn (avg.) |
> > > |-------|-------|-----|
> > > | Qwen3-4B w. default | 22.7 | 83.9 |
> > > | Qwen3-4B w. stronger | 37.2 | 84.8 |
> > > | Qwen3-4B-vanillaRL | 38.0 | 84.8 |
> > > | Qwen3-4B-RL (ours) | 39.4 | 84.8 |
> > > | Qwen2.5-3B-Instruct | / | 72.3 |
> > > | Qwen2.5-3B-Instruct-vanillaRL | / | 78.1 |
> > > | Qwen2.5-3B-Instruct-RL (ours) | / | 78.4 |
> > >
> > > Table 2. Tool-calling performance comparison on BFCL.
> > >
> > > In general, for multi-turn tool-calling, the Qwen3-4B w. stronger baseline is already quite strong. Starting from this high baseline, vanilla RL still provides a modest improvement (37.2 → 38.0), while our method further improves to 39.4. For single-turn,  both vanilla RL and our method substantially improve over the base model, with our method achieving a slightly stronger result (72.3 → 78.1/78.4). Vanilla RL should eventually reach similar final performance, but the focus here is on efficiency. For efficiency, It achieves 1.7x (single turn) and 2.6x (multi-turn) speedup. We are happy to add the comparison of all benchmarks in the final version of the paper. Thank you for your advice.

---

### Official Review · Reviewer_7tjh · 2026-03-12

**Soundness:** 2
**Presentation:** 2
**Significance:** 2
**Originality:** 2
**Overall Recommendation:** 3
**Confidence:** 4

**Summary:**

This paper investigates two complementary aspects of tool-calling in LLM-based agents: (1) the effectiveness of evaluation protocols, and (2) the efficiency of reinforcement learning (RL) training. On the effectiveness side, the authors conduct a systematic sensitivity analysis of the BFCL benchmark, demonstrating that seemingly minor implementation choices—random seeds, multi-turn template construction, reasoning history retention, and system prompt wording—can induce substantial variance in reported performance, particularly in multi-turn settings. On the efficiency side, the paper identifies two computational bottlenecks in RL-based tool-calling training: (i) a high fraction of "zero-variance" prompts that yield no learning signal during rollouts, and (ii) disproportionate wall-clock cost during policy updates due to long sequence lengths. To address these, the authors propose two techniques: online pre-rollout filtering (skipping prompts that have been consistently solved) and variance-aware rollout down-sampling (backpropagating only on rollouts with maximal reward variance). Empirical results on Qwen-series models report meaningful wall-clock speedups without degrading final performance on BFCL and ACEBench.

**Compliance With Llm Reviewing Policy:**

Affirmed.

**Final Justification:**

While the rebuttal resolved some of my issues, the core problem remains unaddressed (see the rebuttal ack). I am maintaining my original score.

**Key Questions For Authors:**

See in "weakness" section

**Limitations:**

The authors include an "Impact Statement" and briefly acknowledge some limitations, but the discussion lacks depth and specificity.

**Strengths And Weaknesses:**

### Strengths:
1. The empirical analysis of evaluation sensitivity (random seeds, template construction, history handling, system prompts) is methodologically careful and supported by controlled experiments across multiple models.


### Weaknesses:
1. Temporal stability assumption lacks theoretical grounding: The pre-rollout filtering method assumes that prompts which are "all-correct" for k consecutive epochs will remain so, but no formal analysis or robustness guarantees are provided. The empirical evidence (Fig. 5) is limited to the specific training setting and may not generalize to distribution shift or heterogeneous data.
2. Potential gradient bias from down-sampling: Selecting only extreme-reward rollouts for backpropagation may introduce systematic bias in advantage estimation, particularly in sparse-reward or multi-objective settings. The paper does not analyze the impact on convergence properties or effective sample size.
3. Limited ablation and interaction analysis: The paper does not isolate the individual contribution of each efficiency technique, nor does it analyze whether their benefits are additive or redundant. This makes it difficult to assess which component drives the reported gains.
4. Evaluation scope is narrow: All efficiency experiments are conducted on relatively small models (3B–4B) using GRPO within the VERL framework. The generalizability to other RL algorithms (PPO, DAPO), larger scales (>70B), or distributed training settings is untested.
5. Lack of novel theoretical or algorithmic insight: The efficiency techniques are straightforward adaptations of existing ideas. The paper does not introduce new optimization principles, convergence analyses, or generalizable design patterns that would influence future work beyond tool-calling.

---

> ### Author Rebuttal · Authors · 2026-03-31
>
> The authors thank the reviewer for your review and constructive feedback. We provide detailed replies below:
>
> **Q1**: Lack formal analysis of temporal stability and convergence properties of down-sampling
> **A1**: Thank you for this comment. We note that this work is primarily empirical, focusing on understanding practical tool-calling issues of evaluation and training. Methods to improve training efficiency are partly inspired by prior work and grounded in empirical observations of training dynamics on both math reasoning and agentic tool-calling scenarios. We agree that a more formal theoretical mathematical analysis would be valuable; however, providing rigorous theoretical guarantees extends beyond the scope of this empirical paper.
>
> **Q2**: Evaluation scope is narrow: All efficiency experiments are conducted on relatively small models (3B–4B) using GRPO within the VERL framework. The generalizability to other RL algorithms (PPO, DAPO), larger scales (>70B), or distributed training settings is untested.
> **A2**: Thank you for this comment. We agree that our efficiency experiments are conducted in a relatively focused setting.
> Regarding **PPO**, prior work (e.g., [1,2]) has shown that GRPO is more effective in tool-calling RL settings, thus we only focus on GRPO. More importantly, the line of zero-variance prompt bottleneck is tied to group-based normalization in GRPO-style objectives, which motivates our focus on this framework.
> Regarding **DAPO**, we observed in early experimentation that DAPO significantly increases total training time due to requiring much more rollout overhead. This is also aligned with observations reported from other papers, e.g., “DS incurs substantial overhead from extensive LLM generation on enlarged batches, which in practice often outweighs the cost of finetuning itself” [3].
> For very **large-scale** models (>70B) with distributed training settings on tool-calling (with quite long system prompts), such experiments typically require at least 6-8 nodes (each with 4 A100 GPUs). This level of computational demand is **extremely expensive** and typically **beyond** the scope of standard academic setups.
>
> [1]. ToolRL: Reward is All Tool Learning Needs, NeurIPS 2025
> [2]. Nemotron-Research-Tool-N1: Exploring Tool-Using Language Models with Reinforced Reasoning, ICLR 2026
> [3]. Dynamics-Predictive Sampling for Active RL Finetuning of Large Reasoning Models, ICLR 2026
>
> **Q3**: The paper does not introduce new optimization principles, convergence analyses, or generalizable design patterns that would influence future work beyond tool-calling.
> **A3**: We clarify that the goal of this work is not to introduce a new optimization algorithm, but to provide a systematic empirical study of evaluation and practical efficiency issues **specifically in tool-calling**, which we find to be underexplored despite its extremely practical importance to the ML research community and practitioners.
> We believe the main contribution lies in identifying practically important bottlenecks, quantifying their impact, and turning them into concrete recommendations for future tool-calling research and evaluation practice.

---

> > ### Author Rebuttal · Reviewer_7tjh · 2026-04-03
> >
> > Thank you for your response. However, I believe my primary concerns remain unaddressed. First, many of the training techniques proposed in this paper appear to be model-specific. The observed phenomena may stem from the inherent capacity limitations of small-scale models rather than being generalizable characteristics. Second, strategies such as skipping correct rollouts are already widely utilized in reinforcement learning (e.g., DAPO). Consequently, I maintain my original assessment.

---

> > > ### Author Response · Authors · 2026-04-07
> > >
> > > We thank the reviewer for the further comments. We provide detailed replies below:
> > >
> > > **Q1**: Many of the training techniques proposed in this paper appear to be model-specific. The observed phenomena may stem from the inherent capacity limitations of small-scale models rather than being generalizable characteristics.
> > > **A1**: We thank the reviewer for this question. To evaluate whether the observed phenomena are model-specific, we extend our experiments to **larger-scale** models, including Llama3.1-8B, Qwen2.5-7B and Qwen3-8B.
> > > (1). As shown in the top figure [here](https://anonymous.4open.science/r/Anonymity3-6D10/README.md), we observe the same pattern of a high zero-variance prompt ratio in tool-calling RL. Notably, this issue becomes more pronounced as model size increases.
> > > (2). The bottom figure shows the policy update phase continues to dominate the wall-clock training time for the 8B model.
> > > Overall, this suggests that the observed phenomena is not a consequence of limited model capacity, but a more general characteristic of the tool-calling RL.
> > >
> > > Besides, regarding which **individual technique** drives the reported gains, for single-turn, the policy update overhead is not dominant, so we **only apply** the first technique; the observed efficiency gains therefore come only from it. For multi-turn, the second technique drives the benefit. We are happy to give more detailed discussion and insights in the final version of the paper. Thank you for your advice.
> > >
> > >
> > > **Q2**: Strategies such as skipping correct rollouts are already widely utilized in reinforcement learning (e.g., DAPO)
> > > **A2**: Thanks for this question. We would like to clarify that our method differs fundamentally from DAPO in when the filtering is applied.
> > > DAPO (i.e., dynamic sampling): **post-rollout** filtering; thus it still performs oversampling in every training step. Even when the model already achieves very high accuracy, it still incurs the **full** rollout generation **cost** before discarding uninformative samples.
> > > Ours: **pre-rollout** filtering; filter at the dataset level to directly reduce unnecessary rollout generation.
> > >
> > > Empirical evidence: we report the averaged wall-clock training time per training step for Qwen2.5-3B-Instruct on single-turn tool-calling:
> > >
> > > | Method | Total rollout generation (s) | Policy update (s) |
> > > |-----|-----|----|
> > > | GRPO | 68.1 | 44.2 |
> > > | Dynamic sampling | 174.1 | 42.0 |
> > > | Online pre-rollout | 75.4 | 44.9 |
> > >
> > > Dynamic sampling significantly increases rollout time (≈2.5× compared to GRPO),  due to mandatory oversampling at every step. Our lightweight cache introduces only marginal overhead on rollout generation while avoiding redundant rollout generation.

---

### Official Review · Reviewer_hSMD · 2026-03-13

**Soundness:** 2
**Presentation:** 4
**Significance:** 3
**Originality:** 2
**Overall Recommendation:** 4
**Confidence:** 4

**Summary:**

This paper studies design questions in how tool-calling agents are measured and trained, using the BFCL tool-calling benchmark for evaluation. For example, analysis on measurement includes experiments showing sensitivity of BFCL performance to random seed variance, changes chat templates and prompts. For training, the paper focuses on two simple optimizations to help reduce RL training time: (1) dynamically filtering out prompts that are likely to provide no training signal/reward variance based on past rollout performance on these prompts; and (2) down-sampling some rollouts before the policy update.

**Compliance With Llm Reviewing Policy:**

Affirmed.

**Final Justification:**

I maintain my original score since my concerns were not fully addressed (as discussed in the rebuttal thread).

**Key Questions For Authors:**

Please see weaknesses above.

**Limitations:**

Yes

**Strengths And Weaknesses:**

Strengths:
1. The recipe and ablations performed for takeaways on what matters for tool-calling agent performance are helpful to reduce design search space for AI practictioners and researchers who want to build/train tool-calling agents.
2. The computational saving methods for RL training proposed are simple to implement, intuitive and practical.


Weaknesses:
1. Only one tool-calling benchmark (BFCL) was used for the experiments and takeaways from the paper. It is unclear if some of the findings would hold on other more complex tool-calling/agent benchmarks (e.g., AppWorld, TerminalBench, WebArena, etc.)

2. The down-sampling of rollouts to reduce policy update wall-clock time is somewhat wasteful as it throws away data that was already was produced with rollout computation. The policy update overhead can be reduced through other ways without throwing away data (e.g., overlapping rollouts and policy updates with Async/Pipelined RL). It would be useful to compare the performance of these approaches.

---

> ### Author Rebuttal · Authors · 2026-03-31
>
> We thank the reviewer for recognizing the helpfulness of takeaways and the intuitiveness and practicality of computational saving methods!
> **Q1**: Only one tool-calling benchmark (BFCL) was used for the experiments and takeaways from the paper. It is unclear if some of the findings would hold on other more complex tool-calling/agent benchmarks (e.g., AppWorld, TerminalBench, WebArena, etc.)
> **A1**: Thank you for this question. In this paper, we focus on general tool-use capabilities and therefore choose BFCL as a representative case study. It’s still widely used by many papers and technical reports. BFCL covers a diverse set of domains and function-calling scenarios including coding, terminal, and tool interactions. As such, we expect our findings, particularly those related to evaluation sensitivity and training efficiency, to **generalize** to other agent benchmarks.
> Some evidence from existing papers: In an Android coding benchmark paper [1], authors perform seed-variance evaluation and find “Agent performance **varies** much more using different **seeds.**”
> [1]. AndroidWorld: A Dynamic Benchmarking Environment for Autonomous Agents, ICLR 2025
>
> **Q2**: Differences with approaches such as Async/Pipelined RL
> **A2**: Thanks for this comment. We agree that methods such as Async/Pipelined RL reduce policy update overhead from the **system-level** by decoupling rollout generation and training.  This reflects a distinct motivation compared to our approach, where we reduce the impact of non-informative rollouts by selectively using more informative samples. Async/Pipelined RL typically requires **modifying** the **whole** training pipeline, while our method is **lightweight** and can be easily integrated into existing frameworks by modifying several lines of code. As such, these methods belong to different lines of work and are largely complementary.

---

> > ### Author Rebuttal · Reviewer_hSMD · 2026-04-03
> >
> > Thanks for your response. I will keep my current scores as my concerns have not been fully resolved. (1) While BFCL is one case study, empirical evidence on just one benchmark does not provide enough confidence that the results will hold more generally. (2) I disagree that Async/Pipelined RL requires modifying the whole training pipeline as this has become a fairly standard and readily available feature of many popular open-source RL frameworks (AReaL, SkyRL, PrimeRL, Slime, etc. -- just to name a few). Without additional evidence, it is not clear to me that the down-sampling approach discussed in the paper provides significant benefits.

---

> > > ### Author Response · Authors · 2026-04-07
> > >
> > > Thanks for the reply and constructive feedback. We agree with the reviewer that the orthogonal method Async/Pipelined RL becomes a famous practical technique nowadays. However, our implementation is built on VERL, which did not natively support Async/Pipelined RL. Incorporating such methods would require non-trivial modifications to the existing training pipeline.

---

### Official Review · Reviewer_xTTW · 2026-03-13

**Soundness:** 2
**Presentation:** 3
**Significance:** 2
**Originality:** 2
**Overall Recommendation:** 4
**Confidence:** 3

**Summary:**

This paper studies tool-calling for LLM agents along two axes. On effectiveness, it systematically probes the BFCL evaluation pipeline across five sensitivity dimensions: random seeds, multi-turn template construction (native vs. context serialization), thinking-history retention, system prompts, and training-data format. The analysis reveals that seemingly minor, often unreported choices can shift multi-turn scores by 3-8%, and that a hand-tuned system prompt alone can match or exceed RL fine-tuning gains (Takeaway 3.4). On efficiency, the paper identifies two sources of waste in GRPO-based tool-calling RL: roughly 80% of prompts are zero-variance (yielding no gradient signal), and policy updates dominate wall-clock time due to long tool-calling sequences. Two accelerations are proposed: online pre-rollout filtering of consistently-solved prompts and max-variance rollout down-sampling (adopted from Xu et al., 2025b). Experiments on Qwen2.5-3B-Instruct (single-turn) and Qwen3-4B (multi-turn) show improved accuracy-per-GPU-hour, +12.1 overall points on ACEBench, and no downstream degradation on HellaSwag, MMLU, TruthfulQA, or WinoGrande (Table 4).

**Compliance With Llm Reviewing Policy:**

Affirmed.

**Final Justification:**

The evaluation sensitivity analysis across five dimensions is a timely and actionable contribution for the tool-calling community. The rebuttal substantially strengthened the paper: three-seed training results resolved the statistical inconsistency between evaluation rigor and training reporting, and the system-prompt intermediate experiment (separating length from content via a copied-default control) was particularly insightful. The precise speedup factors (1.7x/2.6x) and reasonable arguments for non-comparability with alternative RL methods were also helpful. These responses addressed my main concerns and improved my assessment from 3 to 4.

**Key Questions For Authors:**

1. Can you report three-seed training runs for Table 2 and Figure 7? Given the paper's own argument that single-run results are unreliable, this is essential for credibility.

2. Can you ablate the two efficiency techniques separately? Reporting pre-rollout filtering alone and max-variance down-sampling alone would clarify how much each contributes and whether both are necessary.

3. In the system-prompt experiment, can you quantify the prompt change (e.g., token count before and after) and test intermediate modifications to distinguish the effect of prompt length from prompt content?

4. The training-data experiment (Section 3.5) uses 0.7k examples per condition. Would the conclusions hold at the 2-6k scale used elsewhere in the paper?

**Limitations:**

Partially addressed. The impact statement covers misuse risks. Not discussed: the tension between advocating multi-seed evaluation and reporting single-run training, the narrow model coverage for efficiency claims, and the borrowed nature of both acceleration techniques.

**Strengths And Weaknesses:**

The evaluation sensitivity analysis is the paper's clear strength. Examining five axes of variation in a single, controlled study is valuable, and the findings are actionable. The template construction experiment (native vs. context, ~6-8% gap on BFCL multi-turn, Figure 3 left) and the system-prompt finding (Takeaway 3.4, where prompt engineering rivals RL training for Qwen3-4B) are both striking and practically important for anyone building or benchmarking tool-calling agents. The pre-rollout filtering idea is well-motivated by the temporal stability analysis in Figure 5, and the downstream-task check in Table 4 is a responsible addition.

Several weaknesses limit the contribution:

**The paper's own statistical standards are applied inconsistently.** Section 3 argues convincingly that single-run reporting is unreliable and adopts three-seed averaging for evaluation. Yet Table 2 explicitly reports RL training results from "a single run," with no error bars for the trained models (the base model row has them). This directly contradicts the paper's central message. If seed sensitivity matters for evaluation, it matters at least as much for training, where stochastic rollout sampling and policy optimization compound variance.

**The system-prompt experiment is presented as a "small, manual modification," but the actual change is substantial.** Comparing the default BFCL prompt (Figure 9, approximately 6 lines) with the "stronger" prompt (Figure 10, roughly 20 lines of detailed multi-turn instructions including error-handling, parameter-checking, and stopping rules), this is closer to prompt engineering than a minor tweak. The mismatch between framing and reality weakens the persuasiveness of Takeaway 3.4, even though the underlying point about prompt sensitivity is valid.

**Both efficiency techniques are borrowed from prior work, and no ablation separates their individual contributions.** The pre-rollout filtering adapts ideas from Zheng et al. (2025) in math reasoning, and the max-variance rollout down-sampling is explicitly adopted from Xu et al. (2025b). Figure 7 reports the two techniques applied jointly, but never in isolation. The paper therefore cannot say which technique drives the observed speedup, nor whether both are necessary for tool-calling specifically.

**The efficiency analysis is narrow: two models, one benchmark family, and no comparison with alternative RL efficiency methods.** Only Qwen2.5-3B-Instruct and Qwen3-4B are trained. The related work section discusses dynamic sampling (Yu et al., 2025), adaptive allocation (Xiong et al., 2025), and entropy-based approaches (Le et al., 2025), but none are compared experimentally. The training-data experiment in Section 3.5 uses unrealistically small 0.7k datasets, limiting the generality of those findings. Figure 7 shows accuracy vs. GPU-hours curves but does not report precise speedup factors, making quantitative comparison difficult.

---

> ### Author Rebuttal · Authors · 2026-03-31
>
> We thank the reviewer for the valuable feedback and recognizing a clear strength of the sensitivity analysis. We provide detailed replies below:
> **Q1**: Can you report three-seed training runs for Table 2 and Figure 7?
> **A1**: Thank you for raising this point. To clarify, we exactly used three-seed training runs in Fig. 7 in the original paper (see the variance bars). Secondly, we add three-seed runs for the last two rows in the following Table 1 and will update the table in the paper accordingly.
>
> |  Qwen3-4B   | BFCL Multi-turn | BFCL Single-turn (Non-live) | BFCL Single-turn (Live) |
> |----|----|----|----|
> | Base    | 22.7 ± 0.9 | 88.5 ± 0.6 | 79.3 ± 0.4 |
> | Training on single-turn  | 20.2 ± 0.6 | 89.5 ± 0.5 | 79.9 ± 0.2 |
> | Training on multi-turn | 15.9 ± 0.4 | 88.6 ± 0.4 | 80.7 ± 0.1 |
>
> Table 1. Effect of training-data format on Qwen3-4B.
>
> **Q2**: For system-prompt, quantify the prompt change and test intermediate modifications
> **A2**: We appreciate this question. First, we report **both** interpretations of “token count” for clarity.
>
> If referring to **output** prompt length, we present the token count changes in the figure [here](https://anonymous.4open.science/r/Anonymity1-B3A6/README.md), where it counts the token count distribution of default and stronger sys prompt for multi-turn base category. From the figure, although the stronger prompt explicitly encourages multi-turn behavior, it does **not** lead to a noticeable **increase** in reasoning/output token length.
>
> If referring to **input** prompt length, we present the prompt token count changes in the following Table 2.
> To disentangle **prompt length** from **added instructions**, we construct two intermediate variants: (1) duplicating the default prompt once to match length, and (2) a partially strengthened prompt with only instructions 1&2.
> Since multi-turn variance can reach up to 3% (Takeaway 3.1), we report one inference run.
>
> | | default | copying default | stronger (1&2) | stronger (full) |
> |------|----|----|--------|-------|
> | Input token length | 228 | 452 | 302 | 448 |
> | BFCL multi-turn | 22.9  | 23.6 | 36.0 | 37.5 |
>
> Table 2. Influence of input token length and added instructions. BFCL multi-turn for Qwen3-4B with default BFCL prompt, copying prompt, stronger one with 1&2 and full instructions.
>
> Overall, simply increasing input prompt length does not have a positive effect, whereas adding task-relevant instructions yields substantial gains (22.9 → 36.0/37.5), suggesting that the improvement comes from prompt **content** rather than **length** alone.
>
> **Q3**: The training-data experiment (Section 3.5) uses 0.7k examples per condition. Would the conclusions hold at the 2-6k scale used elsewhere in the paper?
> **A3**: Thank you for the question.
>  - Why only 0.7k? It’s due to the limited size of the curated multiturn subsets available from prior datasets (xLAM and ToolACE).
>  - The key finding here is that performance is primarily driven by **trajectory quality** rather than the **data quantity**. Specifically, with higher-quality trajectories (e.g., single-turn here or curated multi-turn in Section 4), training consistently improves performance; with lower-quality trajectories (i.e., multi-turn here), training degrades BFCL performance. We oberve that this trend **persists at larger scales** in early experimentation, as increasing the amount of low-quality trajectories further amplifies error accumulation effects.
>
> **Q4**: Figure 7 shows accuracy vs. GPU-hours curves but does not report precise speedup factors, making quantitative comparison difficult.
> **A4**: Thanks for pointing this out. The efficiency results in Fig. 7 correspond to a precise speedup of **1.7x** (single turn) and **2.6x** (multi-turn).  We will update the paper accordingly.
>
> **Q5**: Comparison with dynamic sampling, adaptive allocation, and entropy-based approaches
> **A5**: We thank the reviewer for this question. We discuss several representative RL efficiency methods and clarify the reasons for not including direct experimental comparisons:
> (1). DAPO (Yu et al., 2025): please refer to the reply to reviewer 7tjh due to space limitation.
> (2). Adaptive allocation (e.g., Xiong et al., 2025): The line of work of selective rollouts actually **increases** per-step training time, e.g., Xiong et al., report a 1.4x-2.8x increase (see Table 2 in their paper) and it does not directly improve end-to-end training efficiency. Thus authors claim it as “a trade-off between computational overhead and performance gain”.
> (3). Entropy-based methods (Le et al., 2025): First, their implementation is **not publicly available**. Moreover, it’s orthogonal to ours rather than directly comparable. Specifically, Le et al. optimize on the entropy of the zero-variance prompts in the current epoch, while our online filtering removes such prompts in subsequent steps/epochs.
>
> We will provide a clarification covering these points in the final version of the paper.

---

> > ### Author Rebuttal · Reviewer_xTTW · 2026-04-04
> >
> > The rebuttal makes sense. I am raising my score to 4.

---

> > > ### Author Response · Authors · 2026-04-07
> > >
> > > We truly appreciate your follow-up and revisiting the assessment. We're glad to have addressed your concern!

---

### Decision · Program_Chairs · 2026-04-30

**Decision:**

Accept (regular)

**Comment:**

In this paper, the authors study tool-calling of models large language models (LLMs) along 2 axes: effectiveness and efficiency. On the former, the authors analyze tool-calling evaluation pipelines and highlight sensitivity to undocumented implementation choices and configurations. On the later, the authors take a close look at RL for tool-calling, identifying issues such as an absence of learning signal and high computational costs. The authors introduce two mechanisms for improving RL-based tool-calling training.

The reviewers highlight various strengths of the paper, including insightful analysis on evaluation robustness and learning efficiency. The reviewers note that the efficiency gains from two methods introduced in the paper are real, intuitive and practical. The reviewers also identified certain limitations of the manuscript, such as single benchmark evaluation, missing theoretical grounding, and limited algorithm comparison. The authors have responded to these during the rebuttal, leading to increases in points and confidence. I think this is an interesting and insightful manuscript that would be of interest to the ICML community. I’m therefore recommending it for acceptance but urge the authors to use the reviewers feedback to improve the paper for the camera-ready.